# Roasted Rye as a Coffee Substitute: Methods for Reducing Acrylamide

**DOI:** 10.3390/foods9070925

**Published:** 2020-07-14

**Authors:** Johannes Pitsch, Otmar Höglinger, Julian Weghuber

**Affiliations:** 1FFoQSI—Austrian Competence Centre for Feed and Food Quality, Safety & Innovation, FFoQSI GmbH, Technopark 1C, 3430 Tulln, Austria; johannes.pitsch@ffoqsi.at (J.P.); otmar.hoeglinger@fh-wels.at (O.H.); 2Department of Food Technology and Nutrition, University of Applied Sciences Upper Austria, 4600 Wels, Austria

**Keywords:** acrylamide, coffee substitute, xanthydrol, roasted rye

## Abstract

Acrylamide is assumed to be a potential carcinogen, and reference values have therefore been implemented in EU legislation. Thus, the food industry needs to reduce the acrylamide content in consumer products to the lowest possible value. In this study, roasted rye was evaluated for its suitability as a coffee substitution product with respect to its acrylamide content. The influence of process modifiers, free asparagine content, storage, and rye type on the final content of acrylamide was investigated. Changes in carbohydrate composition and brightness caused by the roasting process were described. Sample analysis was conducted via GC-MS and HPLC-CAD. Existing methods were adapted to roasted rye as the sample matrix. CaCl_2_ and asparaginase treatment as well as pH adjustments prior to roasting did not prove to reduce the acrylamide content. A significantly (* *p* < 0.027) lower free asparagine content in the raw material resulted in a lower formation of acrylamide in the final product. The acrylamide content significantly decreased (**** *p* < 0.0001) after 3 (1100 ± 18 µg kg^−1^) and 6 (490 ± 7 µg kg^−1^) months of long-term storage. Only samples stored for 6 months (490 ± 7 µg kg^−1^) met the EU acrylamide content requirements (<500 µg kg^−1^) for grain-based coffee substitution products.

## 1. Introduction

Acrylamide is an organic compound and belongs to the chemical group of amides [1]. This compound has been used for decades in polymerized form in biochemistry, in the paper industry, and in wastewater treatment and is therefore produced on an industrial scale by hydrolysis of acrylonitrile [2]. The classification of acrylamide as a “probable carcinogen” in 1994 and its recognition as a possible food contaminant in fried and baked foods in 2002 caused worldwide concern [3,4].

Formed by the Maillard reaction in the presence of reducing sugars and the amino acid asparagine under low-water-content conditions, acrylamide has been of increasing concern for food manufacturers worldwide [5]. The concentration of acrylamide in processed food is strongly dependent on the processing temperature and the composition of raw materials. Food sources rich in proteins and carbohydrates proved to be particularly prone to acrylamide formation, and high amounts of acrylamide were detected in potato chips, French fries [6], bread, coffee [7,8], and cocoa [9,10]. Thus, evaluation of carbohydrate and protein contents (especially that of asparagine) prior and after food processing steps seems to be of importance to determine the relation between the raw materials and final acrylamide content. To inhibit the chemical reactions leading to the formation of acrylamide, the use of NaHCO_3_, CaCl_2_, and asparaginase was discussed [11,12]. Being aware of the acrylamide content and the implementation of measures for its reduction, toxicological studies are necessary to assess the actual risk of this food contaminant.

Hence, model systems have been used to investigate the impact of acrylamide on human health. In the studies conducted, genotoxic and carcinogenic effects were observed [3,13]. To reduce potential health risks, worldwide legislation addressed acrylamide via restrictive orders and process recommendations for the food industry (2013/647/EU; EU-VO 2017/2158; FDA-2013-D-0715; ÖNORM EN 16987). These restrictions affected coffee producers, who were impelled to reduce acrylamide to levels ≤400 µg kg^−1^ for distribution across the EU, and an even lower level of ≤280 µg kg^−1^ was required by the Federal Office of Consumer Protection and Food Safety (BVL) in Germany. These restrictions also applied to companies producing coffee substitution products for people suffering from caffeine intolerance or using them due to ecological considerations. Coffee substitutes can be made of local raw materials, including different grain species, such as *Taraxacum* and *Lupinus*. After roasting, those products can be prepared with hot water, similar to ground coffee beans. As the taste profile of coffee is mainly dominated by roast aromas, a very similar taste can be achieved by targeted roasting of other raw materials [14]. However, those raw materials, like rye, are known to contain high amounts of acrylamide after roasting [15,16]. Therefore, it is of special interest to develop methods for acrylamide reduction in coffee substitutes.

To assess the toxicity of the final product, analytical methods that offer high sensitivity and low limits of detection and quantitation have been developed over the years to reduce effort in sample preparation and to further increase selectivity and cost efficiency [17,18]. Most methods for acrylamide analysis rely on gas and liquid chromatography coupled to mass spectrometry (GC-, LC-MS). However, recent studies attest to the high selectivity and ease of use of other methods such as immunoassays, nanobiosensors, and supramolecular recognition. Most of those novel methods are still under development or evaluation [19]. Therefore, GC and LC methods still represent the gold standard and are therefore applied by many analytical laboratories for acrylamide detection. In addition to acrylamide, quantitation of carbohydrates is strongly suggested, as they can act as reducing agents during food processing and can therefore increase the final acrylamide content. Several techniques are used for carbohydrate determination, i.e., liquid chromatography coupled with refractive index (HPLC-RI) detection or charged aerosol detection (CAD) and high-pressure anion exchange chromatography coupled with pulsed amperometric detection (HPAEC-PAD) [20,21].

For this study, CAD was chosen as the most appropriate option for carbohydrate detection [16]. Advantages over HPLC-RI include a faster separation speed due to gradient elution and lower limits of detection; additionally, sample preparation is faster than that required for HPAEC-PAD. Existing GC-MS methods for acrylamide determination via xanthydrol were adapted to be used for rye sample matrices [22]. The GC-MS method proved to be beneficial in terms of sample preparation speed, and xanthydrol was used as a relatively low-toxicity derivatization reagent. The asparagine content was measured using GC-MS after adaptation of an existing method [23]. We optimized the approach in terms of sample preparation and speed of analysis for asparagine determination in roasted rye. Our aim was to identify a significant relationship between carbohydrate and asparagine contents in roasted rye acrylamide formation and investigate the influence of potential inhibitors on acrylamide reduction.

## 2. Materials and Methods

### 2.1. Chemicals, Rye Samples, and Standards

Standards of acrylamide (≥99.5%), D3-acrylamide (500 mg L^−1^ in acetonitrile), and asparagine (99%), as well as both GC-grade derivatization reagents xanthydrol (≥99%) and N-tert-butyldimethylsilyl-N-methyltrifluoroacetamide (MTBSTFA) (≥99%) with 1% tert-butyldimethylsilyl chloride (TBDMSCI), were obtained from Sigma Aldrich (Schnelldorf, Germany). For pH adjustments, Na_2_CO_3_ and acetic acid (80%) were purchased for VWR (Vienna, Austria).

Analytical-Grade fructose, glucose, maltose, and sucrose were purchased from Sigma Aldrich. LC-MS-grade acetonitrile and ammonium acetate, as well as ammonia (25%), were obtained from VWR. GC-MS grade ethylacetate, hexane, HCl (37% p.A.) and Carrez I and Carrez II solutions were purchased from VWR. Sample cleanup was performed using J.T. Baker BAKERBOND spe C18 polarplus 1 mL SPE (solid phase extraction) columns (Schwerte, DE). LC-MS-grade water (<0.055 μS cm^−1^) from a Sartorius ultrapure water purification system (Göttingen, Germany) was used for sample preparation.

The grain and rye varieties (*Secale cereale*) used were obtained from Arnreiter Mühle GmbH (Wallern a der Trattnach, Austria), Plohberger Malz GmbH (Grieskirchen, Austria), and Saatgut Erntegut GmBH (Linz, Austria). Asparaginase (EC 3.5.1.1, hydrolase) was purchased as PreventAse (product #23156) from DSM (Heerlen, Netherlands), and CaCl_2_ was purchased from Solvay (Brussels, Belgium).

### 2.2. Sample and Standard Preparation

The modifiers asparaginase (100, 200, 300 mg L^−1^), CaCl_2_ (0.1, 0.2 moL L^−1^), Na_2_CO_3_ (pH = 8.0, 8.5, 9.0) and acetic acid (pH = 5.0, 6.0) were diluted in ultrapure water. Grains were soaked in water + modifier for 16 h. Per 400 g sample, 1000 g water + modifier was used. For storage tests, samples were stored at room temperature (25 °C) in their original containers.

#### 2.2.1. Preparation of Internal D3-Acrylamide Standard Solution

D3-acrylamide [D3-AA] solution for use as an internal standard was prepared at 500 µg L^−1^ in water.

#### 2.2.2. Extraction of Acrylamide from Milled Rye Samples

For acrylamide (AA) extraction, 100 mg ± 5 mg of ground sample was weighed into a 2 mL centrifugation tube, and 500 µL of hexane, 900 µL of water and 100 µL of D3-AA solution were added. Extraction was performed at 40 °C for 120 min at 1000 rpm in a thermoshaker. After centrifugation at 17,000× *g* and 25 °C for 5 min, the upper hexane layer was removed. Then, 800 µL of the sample was mixed with 100 µL of Carrez I and 100 µL Carrez II reagent successively. After vortexing for 30 s, samples were centrifuged under the previously mentioned conditions. SPE columns were conditioned according to the manufacturer’s instructions. Further sample cleanup was performed by elution of 500 µL of sample through the SPE column and dilution to 1 mL.

#### 2.2.3. Xanthydrol Derivatization of Extracted Rye Samples (Acrylamide)

Derivatization was performed at 20 °C for 25 min after adding 100 µL of 0.5 moL L^−1^ HCl and 100 µL of 3.5% xanthydrol solution to 800 µL of the diluted sample. Extraction of xanthyl-AA was achieved by adding 500 µL of ethyl acetate to the sample and further shaking under derivatization conditions for 5 min. The sample was centrifuged afterwards under the conditions mentioned above, and 300 µL of the upper ethyl acetate layer was transferred to a 2 mL GC crimp vial for measurement.

#### 2.2.4. Preparation of External and Internal Acrylamide Standards

AA standards without ground samples were prepared separately in triplicate by substitution of 10, 50, 100, 150, 200, 250, 300, 350, and 400 µL of water with a 500 µg L^−1^ AA solution. D3-AA standards without ground samples were prepared separately in triplicate by substitution of 50 µL and 90 µL of the D3-AA solution with water and substitution of 50, 100, 150, 200, 250, and 300 µL of water with a D3-AA solution. Calibration curves were constructed using the peak areas of the obtained 50, 250, 500, 750, 1000, 1250, 1500, 1750, and 2000 µg L^−1^ AA and D3-AA standards.

#### 2.2.5. Preparation of Asparagine Standard Solution

An asparagine solution at 2.5 mg L^−1^ was prepared for use as an external standard.

#### 2.2.6. Extraction of Asparagine from Milled Rye Samples

For asparagine extraction, 100 mg ± 5 mg of ground sample was weighed into a 2 mL centrifugation tube, and 500 µL of hexane, 900 µL of water and 100 µL of a 1.0 moL L^−1^ HCl solution were added. Extraction was performed at 40 °C for 120 min at 1000 rpm in a thermoshaker. After centrifugation at 17,000× *g* and 25 °C for 5 min, the upper hexane layer was removed. SPE columns were conditioned according to the manufacturer’s instructions. Further sample cleanup was performed via elution of 500 µL of the sample through the SPE column and dilution to 10 mL. For deproteinization, 100 µL of the sample was diluted with 900 µL of acetonitrile and centrifuged under the abovementioned conditions. Consecutively, 500 µL of the sample was transferred into a clean centrifugation tube and dried under a stream of nitrogen.

#### 2.2.7. BSTFA Derivatization of Extracted Rye Samples (Asparagine)

After evaporation of solvents, 50 µL of a silylation reagent (MTBSTFA) and 450 µL of acetonitrile were added to the dried tubes. Derivatization was performed at 80 °C for 60 min. After derivatization, the sample was transferred to a 2 mL crimp vial for GC analysis. Asparagine standards were prepared separately without ground samples in triplicate by substitution of 10, 50, 100, 150, 200, 250, 300, 350, and 400 µL of water with a 2.5 mg L^−1^ asparagine solution. Calibration curves were constructed using the peak areas from the silylated 2.5, 5.0, 25.0, 37.5, 50.0, 62.5, 75.0, 87.5, and 100 mg L^−1^ asparagine standards.

#### 2.2.8. Extraction of Carbohydrates from Milled Rye Samples

For carbohydrate extraction, 100 mg ± 5 mg of ground sample was weighed into a 2 mL centrifugation tube, and 1000 µL of water was added. Extraction was performed at 40 °C for 120 min at 1000 rpm in a thermoshaker. After centrifugation at 17,000× *g* and at 25 °C for 5 min, 100 µL of supernatant was transferred into a fresh centrifugation tube and vortexed after adding 900 µL of acetonitrile. Precipitants were removed by repeated centrifugation under the abovementioned conditions. For HPLC measurement, 850 µL of supernatant was transferred into a screw-cap vial. Carbohydrate standards were prepared at concentrations of 0.010, 0.050, 0.10, 0.50, 1.0, and 4.0 g L^−1^ in water. Calibration curves were constructed using the peak areas obtained by measurement. For measurement with charged aerosol detection, no derivatization steps are required.

### 2.3. Instrumentation

Whole grain samples were roasted using a drum roaster TKMSX 1 CAFEMINO (Toper, Izmir, Turkey). Roasted samples were ground using a Perten 3100 (Perten Instruments, Haguenau, France) laboratory mill. Moist grain was dried using a fluid-bed dryer TG100 (Retsch, Haan, Germany).

Color measurements of roasted grain were performed by a VeriColor spectro (x-rite, Grand Rapids, MI, USA) spectrophotometer. Sample brightness is expressed in lightness values (L *) (0 = black; 100 = white).

Derivatizations and extractions were performed using a thermoshaker. Dry-Matter measurements were performed using an MA 35 (Sartorius, Göttingen, Germany) moisture analyzer.

Acrylamide and asparagine sample analyses were performed on a Thermo Trace 1300 GC equipped with a Thermo TSH100 autosampler coupled to a Thermo ISQ 7000 MS (Thermo Fisher Scientific, Waltham, MA, USA). Carbohydrate experiments were performed on an UltiMate 3000 UHPLC system (Thermo Fisher Scientific, Waltham, MA, USA) equipped with a solvent degasser, a quaternary pump, an autosampler, and a thermostatic column compartment and coupled to a Corona Veo instrument (Thermo Fisher Scientific, MA, USA). Data processing was carried out with Chromeleon 7.2.10 software (Thermo Fisher Scientific, MA, USA), and the compressed air gas flow rate was automatically regulated and monitored by the CAD device. Data collection was set to 2.0 Hz at a filter constant of 3.6 s. The power function for response and signal correction was set to 1.30.

Chromatographic separation of acrylamide was achieved using a TRACE TR-5MS (0.25 mm, 0.25 µm, 30 m) column (Thermo Fisher Scientific, Waltham, MA, USA). The injector port temperature was maintained at 250 °C. The GC column temperature was kept at 60 °C for 1 min and increased from 60 °C to 300 °C in 20 min. During measurements, the transfer line was kept at 280 °C, and the ion source was kept at 300 °C. GC was operated with helium (99.999%) at a constant flow rate of 0.8 mL/min. Each sample was determined in triplicate via splitless injection of 1.3 µL of sample. Fragment ions *m/z* = 206/234 and *m/z* = 207/237 were used in selected ion mode for identification of xanthyl-acrylamide and D3-xanthyl-acrylamide, respectively. Quantitation of xanthyl-acrylamide and D3-xanthyl-acrylamide was performed in selected ion mode at *m/z* = 251 and *m/z* = 254, respectively.

Chromatographic separation of asparagine was achieved using a TRACE TR-5MS (0.25 mm, 0.25 µm, 30 m) column (Thermo Fisher Scientific, Waltham, MA, USA). The injector port temperature was kept at 300 °C. The GC column temperature was kept at 60 °C for 1 min and increased from 60 °C to 325 °C in 30 min. During measurements, the transfer line was kept at 280 °C, and the ion source was kept at 300 °C. The GC was operated with helium (99.999%) at a constant flow rate of 1.0 mL/min. Each sample was determined in triplicate via splitless injection of 1.0 µL of sample. The fragment ion *m/z* = 302 was used in selected ion mode for identification of the dimethyl-tert-butylsilyl derivative of asparagine. Quantitation was performed in selected ion mode at *m/z* = 417.

Chromatographic separation for carbohydrates was achieved using a WATERS Acquity UPLC BEH amide column (130 Å, 1.7 µm, 2.1 mm × 150 mm) maintained at 40 °C. The autosampler was kept at 25 °C. A guard column (Acquity UPLC BEH Amide VanGuard precolumn, Milford, MA, USA) was used to protect the column from particles. Mobile phase A was 85% ACN, and mobile phase B was 60% ACN. All mobile phases contained 10 mM NH_4_Ac and were adjusted to pH = 8.25 with NH_4_OH (25%, NH_3_ basis). The buffer components were filtered through a 0.2 µm membrane filter after preparation. The gradient program used is shown in Table 1. The autosampler needle was washed before and after each injection with 100 µL (20 µL s^−1^) of mobile phase A to prevent sample carryover between runs. The injection volume was 5 µL.

## 3. Results

To investigate the applicability of roasted rye a as a coffee substitution product, rye grains of a single type were roasted in a drum roaster. We focused on the following parameters relevant to the formation of acrylamide: (i) filling volume variations in the drum roaster that might also influence the determined concentration of acrylamide; (ii) the asparagine content of rye varieties before and after roasting; (iii) the role of different rye grain types on the formation of acrylamide; (iv) the roasting temperature; (v) modifiers (CaCl_2_, asparaginase, and pH adjustments), as they potentially impact acrylamide formation; (vi) long-term storage of products at room temperature to validate its influence on acrylamide stability; (vii) the roasting temperature-dependent change in rye flour brightness; and (viii) carbohydrate content and composition before and after roasting.

### 3.1. Filling Volume

According to the manufacturer’s specifications, the drum roaster used in this study accommodates approximately 1000 g of coffee beans for roasting in one pass. Due to variations in the density of the raw material, it was chosen to test sample weights of 500 ± 200 g for maximum flexibility. The results of acrylamide formation as a function of filling volume are depicted in Figure 1. Volumes of 700 g (1182 ± 12 µg kg^−1^) and 300 g (1249 ± 15 µg kg^−1^) showed the lowest standard deviations of all tested filling volumes. No significant impact of filling volume on the formation of acrylamide could be stated. As 400 g of rye per sample was sufficient for analysis, this volume was considered for further experiments.

Regardless of the degree of filling, the rye was roasted for 13 minutes at a constant air temperature of 190 °C. During the roasting process, it was not possible to measure temperature of the individual grains. Therefore, grain brightness was evaluated after roasting (Table 2).

Larger sample volumes were brighter and showed lower acrylamide values than smaller volumes. As the drum roasters heating power was limited, larger filling volumes required more time to heat up compared to smaller volumes and it can be assumed that average grain temperature was lower along the roasting process, respectively.

### 3.2. Asparagine Content of Samples before and after Roasting

All four rye varieties used in this study as well as samples of commercially available wheat, spelt, and barley were analyzed for their asparagine content prior to roasting (Figure 2). Rye variety 4 showed the highest asparagine content (658 ± 32 mg kg^−1^) in this study. Only rye variety 1 had a significantly (* *p* < 0.016) lower asparagine content (560 ± 22 mg kg^−1^) than the reference samples. Based on the calibration standards, the limit of quantitation (LOQ) was 25 mg kg^-1^ for the GC method used. None of the roasted rye samples contained asparagine within the range of calibration (<2.5 mg L^−1^).

### 3.3. Rye Types

Next, the influence of four commercially available rye types on acrylamide formation was tested. Rye variety 4 showed the highest tendency for acrylamide formation (1211 ± 28 µg kg^−1^), as depicted in Figure 3.

One of the aims of this work was to test for inhibition factors of acrylamide formation. Therefore, rye type 4 was chosen for further experiments, as a high initial acrylamide content could improve the sensitivity of further experiments regarding optimized roasting parameters or modifiers.

### 3.4. Roasting Temperature

For evaluation of the roasting temperature on acrylamide formation, 400 g of rye type 4 was roasted at five different temperatures typical for coffee bean roasting.

The results depicted in Figure 4 indicate the lowest acrylamide content in samples roasted at 120 °C (640 ± 15 µg kg^−1^). The highest acrylamide content (1700 ± 26 µg kg^−1^) was observed for samples roasted at 150 °C. The water content after the roasting process ranged from 1.85–2.39%.

### 3.5. Modifiers for Inhibition of Acrylamide Formation

The role of modifiers in acrylamide formation was investigated. The results for asparaginase-treated samples (Figure 5a) were compared to an untreated sample (control). After 20 min of roasting, the acrylamide content increased with low significance (* *p* < 0.05) in samples treated with 300 mg∙L^−1^ asparaginase (1000 ± 12 µg kg^−1^). Untreated (890 ± 35 µg kg^−1^) and 200 mg∙L^−1^ (890 ± 16 µg kg^−1^) samples showed almost equal acrylamide values. As a control experiment, the acrylamide content was determined at reduced roasting times. As expected, samples treated with 100 mg∙L^−1^ asparaginase showed an increased acrylamide content with increased roasting time.

Samples treated with CaCl_2_ (Figure 5b) showed a significantly increased (* *p* < 0.028) acrylamide content compared to that of the control sample. During roasting, the acrylamide content first increased (15 min; 1287 ± 23 µg kg^−1^) and then decreased (20 min; 980 ± 23 µg kg^−1^) in samples treated with 0.2 moL L^−1^ CaCl_2_. The water content after the roasting process ranged from 0.17–4.02%.

Samples soaked in pH-adjusted water (Figure 5c) were compared to untreated samples after roasting. All pH-adjusted samples showed a significant increase in acrylamide content compared to that of the control sample (890 ± 35 µg kg^−1^). In samples adjusted to pH = 8.0 (1000 ± 17 µg kg^−1^) and pH = 9.0 (940 ± 18 µg kg^−1^), the increase in acrylamide content was lower than that in acidic samples adjusted to pH = 5.0 (1182 ± 28 µg kg^−1^) and pH = 6.0 (1127 ± 11 µg kg^−1^). The acidic samples seemed to be more prone to acrylamide formation than the alkaline samples.

### 3.6. Degradation of Acrylamide after Controlled Storage

The influence of storage duration on acrylamide content in roasted rye was tested. Degradation of acrylamide (Figure 6) proved to be highly significant (**** *p* < 0.0001) after 3 (1100 ± 18 µg kg^−1^) and 6 months (490 ± 7 µg kg^−1^) of storage compared to that acrylamide content in the fresh samples (1400 ± 11 µg kg^−1^). The water content increased from 1.85–2.59% to 2.41–3.12% over 6 months.

### 3.7. Impact of Temperature on Sample Brightness

The brightness decrease caused by roasting (*R*^2^ = 0.9194) was observed for all samples in Figure 7. The decrease in brightness showed a linear correlation with temperature in the range from 120 °C to 200 °C. Deviations in sample brightness were related mainly to deviances in dry matter content of the samples before roasting. A relatively high initial sample humidity led to a small reduction in brightness. Prolonged roasting did not further change the brightness value, as residual moisture in the samples remained at 1.85 ± 0.24%.

### 3.8. Carbohydrate Content of Roasted Rye

The contents of fructose, glucose, sucrose, and maltose in rye flour before and after roasting were determined (Figure 8). The maltose (5.43 ± 0.333 g kg^−1^) and fructose (0.44 ± 0.004 g kg^−1^) contents increased (* *p* < 0.0145; ** *p* < 0.004), whereas the glucose (0.83 ± 0.007 g kg^−1^) and sucrose (0.33 ± 0.065 g kg^−1^) contents decreased (* *p* < 0.042; *** *p* < 0.0006) after roasting.

The different batches showed the highest variation in maltose content (27%) and the lowest variation in sucrose content (7%). The content of reducing carbohydrates involved in acrylamide formation, fructose (10%), and glucose (13%), varied remarkably between rye batches. The results indicate that maltose can be formed by heat treatment of glucose, whereas fructose is released from polymers or formed from sucrose.

## 4. Discussion

In this study, we investigated the efficacy of modifiers, substances that were added before or during the roasting process, to inhibit acrylamide formation. Therefore, a rye variety with a high initial acrylamide content after roasting (rye variety 4) was chosen. We found that the free asparagine content (Figure 2) in the raw material correlated with acrylamide formation in the roasting process (Figure 3), which was consistent with previous studies [3,24]. Accordingly, the use of rye with a low asparagine content for roasting should effectively reduce acrylamide in the final product. However, the asparagine content variation in different types of grain was low and could therefore not be considered the sole parameter for effective acrylamide reduction. To reduce the asparagine content in the raw material, asparaginase was used at concentrations suggested in the literature [11]. The pretreatment of rye grains with asparaginase was ineffective at reducing the final acrylamide concentration (Figure 5a). However, a prolonged roasting time decreased the acrylamide content at the chosen roasting temperature (190 °C), potentially caused by decomposition and volatilization of acrylamide, consistent with other studies [25]. Untreated rye grains and treated rye grains showed comparable acrylamide contents in their final product after 20 min of roasting. It is possible that the grains did not absorb enough asparaginase to effectively reduce the free asparagine content. Therefore, mashed grains were analyzed after asparaginase treatment. Importantly, we also did not detect a reduction in the free asparagine content under these conditions.

We also studied the influence of the roasting temperature (Figure 4) on acrylamide formation. The obtained results clearly showed that acrylamide formation at 120 °C was lower than that at temperatures above 140 °C, which was consistent with other studies [26,27]. However, the acrylamide content of samples roasted at 120 °C did not meet the suggested reference values (EU-VO 2017/2158). Acrylamide formation at 140–170 °C was much higher than that at lower temperatures, although no linear temperature correlation was observed.

CaCl_2_, acetic acid, and Na_2_CO_3_ were tested as modifiers and compared to results from other studies performed on different kinds of foodstuffs [28]. CaCl_2_ had no effect on acrylamide content in the final product (Figure 5b) but led to a pronounced bitter taste that would possibly be refused by costumers. Acetic acid and Na_2_CO_3_ (Figure 5c) were used for pH adjustment and did not affect the taste of the final product. However, pH adjustment did not reduce the acrylamide content compared to that of unadjusted samples. For acidic foodstuffs, which naturally exhibit pH values <7.0 and need to be roasted, adjustment to pH = 7.0 appeared beneficial for acrylamide reduction, as acidic samples showed significantly higher acrylamide contents. While it is known that acidification can reduce acrylamide content in various foods including potato crisps and French fries [29,30], we did not observe a similar effect for rye grains. We speculate that acetic acid, which was used for acidification, was lost during the roasting process due to its volatility. This could potentially explain increased acrylamide values of the acidified product.

Storage of roasted samples for a duration of 3 and 6 months showed a highly significant reduction in acrylamide content (Figure 6), consistent with a previous study [31]. According to Hoenicke et al., acrylamide remains stable in liquid beverages but is reduced in powdered foodstuffs. We could confirm these results by controlled storage tests in this study for powdered coffee substitution products. Although storage duration of six months did not meet the reference values of the EU, longer storage could be beneficial in the reduction of acrylamide content in the final product. Further evaluation of the impact of storage on the taste and other sensorial properties appears necessary to render this acrylamide reduction approach suitable for product treatment.

The influence of roasting temperature on the brightness value (L *) showed a linear correlation (Figure 7). Stabilization of sample brightness in the temperature range between 170 °C and 190 °C was observable. Reasonably, this is a result of pyrolysis and the release of a significant amount of water, which limits the roasting process. With increasing temperature at a constant roasting duration, darkening of samples was detectable. This effect may be substantially caused by the Maillard reaction. In addition, other, not further specified chemical reactions in rye grains may contribute. In this context, alterations in the chemical composition, such as polymerization reactions, induced by the roasting process can affect the macronutrient content. Therefore, we analyzed the carbohydrate content in addition to acrylamide and asparagine quantitation. Acrylamide formation is dependent on the content of reducing sugars present in the sample [32,33], but carbohydrates also influence the flavor profile and palate of foodstuffs. Alterations of the carbohydrate content in rye cannot be achieved without huge effort, i.e., washing steps or extraction procedures, and is presumably not an option. Our study indicates (Figure 8) that reduced glucose and sucrose contents, in contrast to moderately increased fructose and extensively increased maltose contents, results in reduced sweetness and increased mouthfeel. This effect is desired by many coffee consumers and renders roasted rye a suitable coffee substitution.

Importantly, consumer acceptance is strongly correlated with the brightness value, as higher roasting temperatures increase typical roast flavors demanded by the customer. Rye roasted at temperatures above 170 °C meets the requirements of brightness values <65. Therefore, future studies should focus on the reduction of acrylamide content in rye roasted above those temperatures to further increase consumer acceptance.

## 5. Conclusions

The acrylamide content in roasted rye was higher than that in roasted coffee beans. The modifiers used in this study did not decrease acrylamide formation. However, long-term storage and low-temperature roasting can be used to reduce the acrylamide content in the final product. The brightness of the product is directly related to the roasting temperature, which also changes the carbohydrate profile.

## Figures and Tables

**Figure 1 foods-09-00925-f001:**
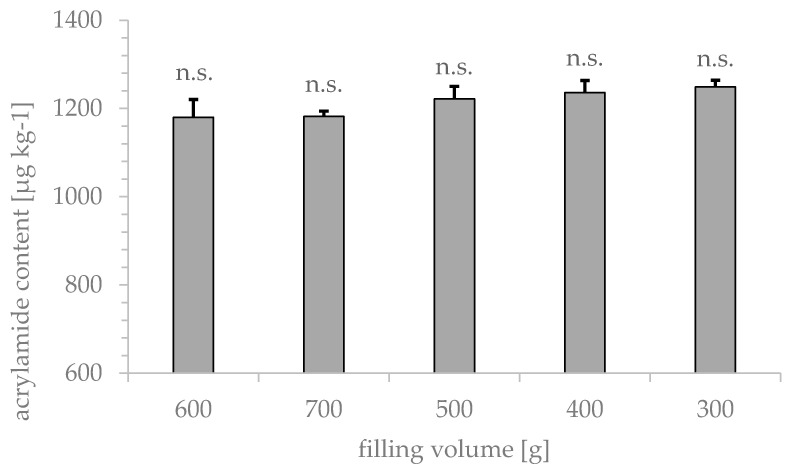
Impact of rye grain filling volume in the drum roaster on acrylamide content measured by GC-MS after xanthydrol derivatization. Whole grain samples (rye variety 4) were roasted at 190 °C for 13 min (*n* = 5).

**Figure 2 foods-09-00925-f002:**
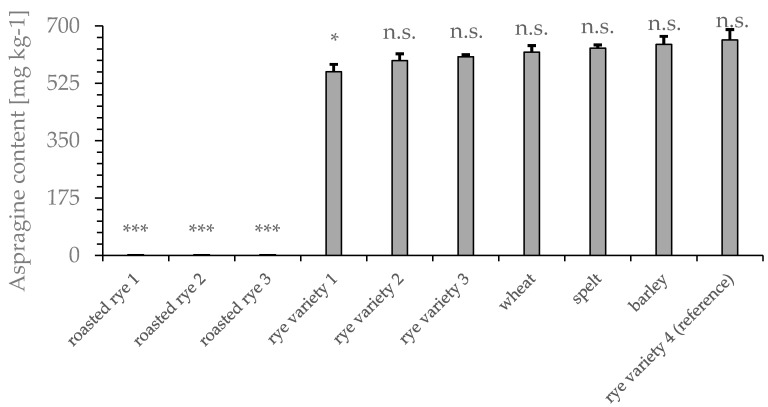
Free asparagine content of four rye varieties, wheat, spelt, and barley, analyzed by GC-MS after MTBSTFA derivatization. Roasted rye sample 1: 150 °C, 20 min; sample 2: 170 °C, 20 min; sample 3: 190 °C, 20 min; sample 4 (reference): 150 °C, 20 min. *n* = 6, *** *p* < 0.001 and * *p* < 0.05 indicate significant differences from the reference.

**Figure 3 foods-09-00925-f003:**
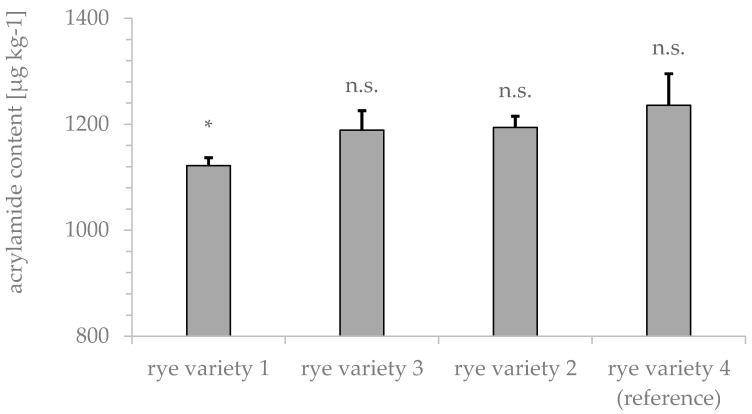
Acrylamide content of four different rye types analyzed by GC-MS after xanthydrol derivatization. Samples (*n* = 5) were roasted at 185 °C for 20 min. * *p* < 0.05 indicates significant differences from the reference.

**Figure 4 foods-09-00925-f004:**
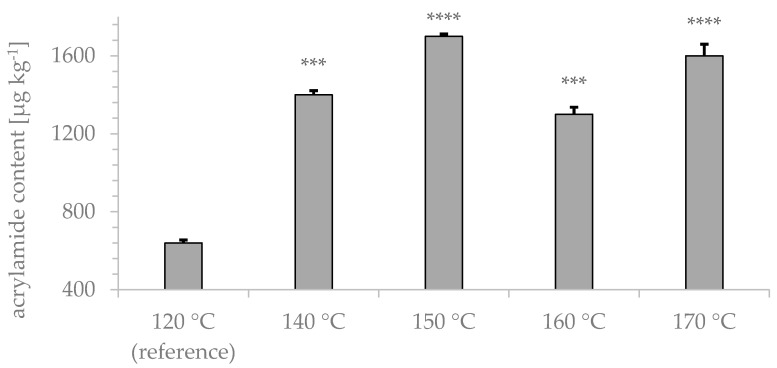
Influence of roasting temperature on acrylamide formation. Samples (*n* = 5) roasted at 120 °C, 140 °C, 150 °C, 160 °C, and 170 °C for 45 min were analyzed by GC-MS after xanthydrol derivatization. **** *p* < 0.0001, *** *p* < 0.001, ** *p* < 0.01 and * *p* < 0.05 indicate significant differences from the reference.

**Figure 5 foods-09-00925-f005:**
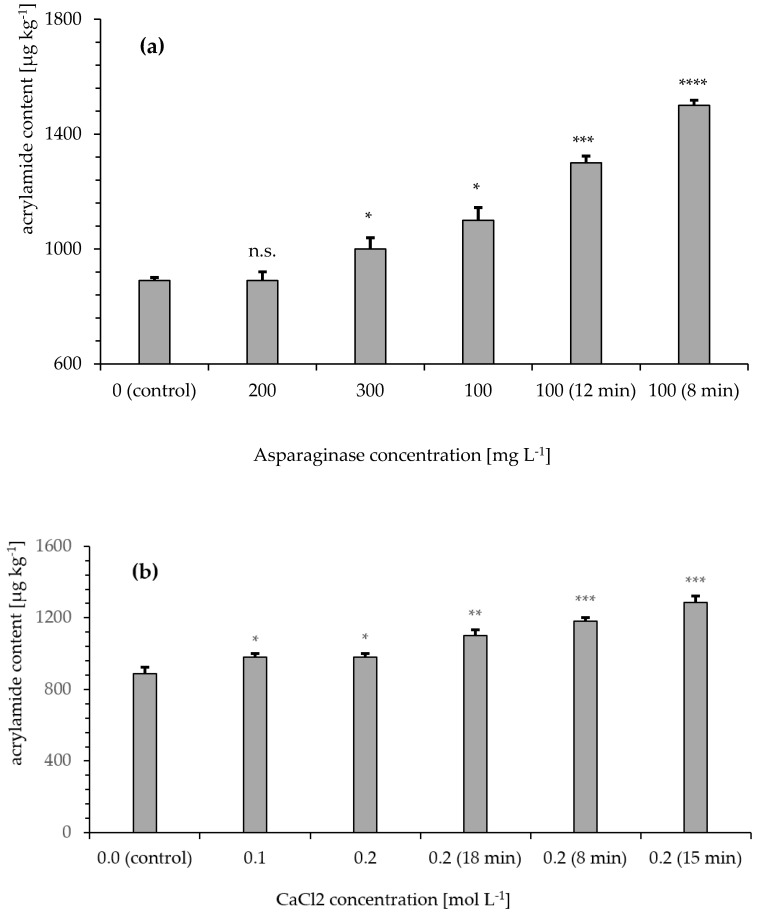
Acrylamide content of roasted and milled rye after treatment; analyzed by GC-MS after xanthydrol derivatization. Whole grain samples (*n* = 5) treated with a modifier solution for 16 h at 25 °C, washed with water, and dried prior to roasting. **** *p* < 0.0001, *** *p* < 0.001, ** *p* < 0.01 and * *p* < 0.05 indicate significant differences from the control. Samples were roasted for 20 min, unless otherwise stated in brackets. (**a**) Asparaginase (asparaginase concentration [mg∙L^−1^]); roasting at 190 °C. (**b**) CaCl_2_ (CaCl_2_ concentration [moL L^−1^]); roasting at 185 °C. (**c**) pH adjustment with acetic acid and Na_2_CO_3_ (pH value); roasting at 185 °C.

**Figure 6 foods-09-00925-f006:**
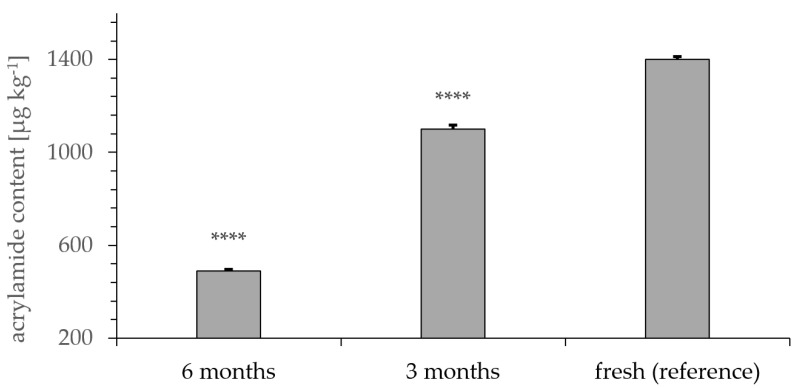
Acrylamide content of roasted and milled rye analyzed by GC-MS after xanthydrol derivatization. Samples were measured fresh and after 3 months and 6 months of controlled storage. Samples (*n* = 5) were roasted at 140 °C for 30 min and stored in closed plastic containers at 25 °C. **** *p* < 0.0001 indicates significant differences from the reference.

**Figure 7 foods-09-00925-f007:**
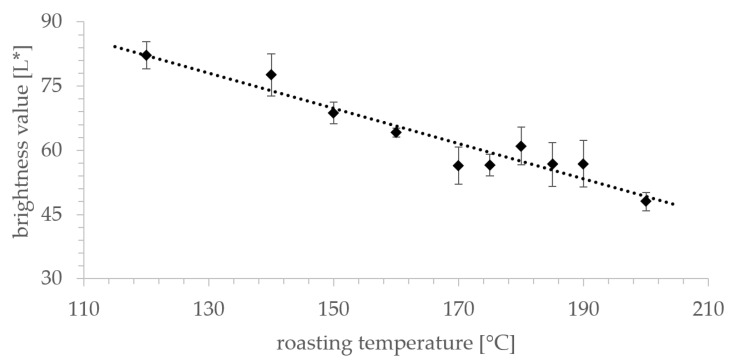
Brightness values (L *) of roasted rye; samples (*n* = 5) were measured with an X-rite VeriColor spectro spectrophotometer. Roasting duration: 20 min.

**Figure 8 foods-09-00925-f008:**
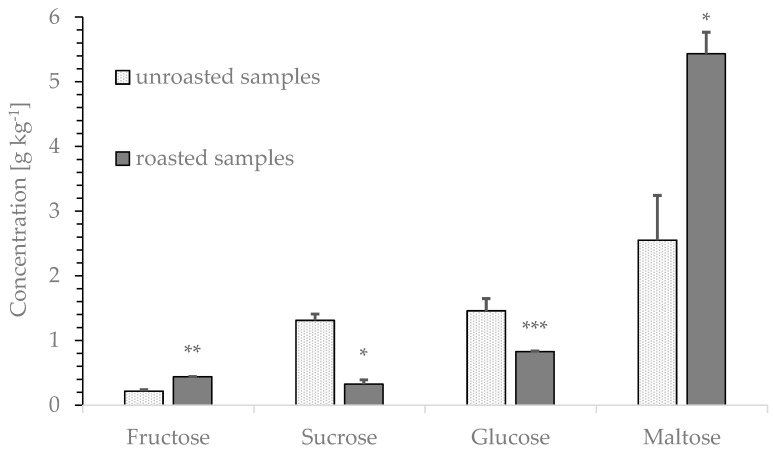
Carbohydrate content of rye flour and roasted rye analyzed by HPLC-CAD. Rye variety 4 milled to flour; rye variety 4 roasted at 170 °C for 20 min. *n* = 5 for all samples; *** *p* < 0.001, ** *p* < 0.01 and * *p* < 0.05 indicate significant differences from the unroasted samples.

**Table 1 foods-09-00925-t001:** Solvent gradient elution program used for the elution of sugars. Negative values represent equilibration times.

Time (min)	%A	%B	Flow Rate (mL min^−1^)
−35.000	100.00	0.000	0.300
−30.100	100.00	0.000	0.300
−30.000	100.00	0.000	0.150
−11.000	100.00	0.000	0.150
−10.000	100.00	0.000	0.300
0.000	100.00	0.000	0.300
10.000	100.00	0.000	0.300
35.000	0.000	100.00	0.300
45.000	0.000	100.00	0.300

Statistical analysis, i.e., linear regression and intra- and interday repeatability, was performed with Chromeleon 7.2.8 software and Microsoft Office Excel 365 (Redmond, WA, USA).

**Table 2 foods-09-00925-t002:** Brightness (L *) of rye grains after roasting.

Rye [g]	L *
600	59.37
700	58.21
500	55.27
400	54.13
300	50.12

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
