# Peer review of "Roasted Rye as a Coffee Substitute: Methods for Reducing Acrylamide"

_foods, 2020, doi:10.3390/foods9070925_

Round 1
Reviewer 1 Report
The manuscript describes the usefulness of roasted rye as a substitute for roasted coffee. Such an idea resulted from the market demand for coffee-like products due to the needs of consumers who do not want to consume caffeine. Currently, grains of other cereals are used for this purpose, e.g. of barley or wheat, roasted chicory is also used. Currently used roasted substitutes have a high content of acrylamide, whose presence in food should be limited. Hence the search for other substitutes.
The study uses rye that has undergone various modifications before or during roasting. None of the modifications allowed to achieve the level of acrylamide lower than recommended in the EU. Even if we notice that the authors chose for the study a variety especially predisposed to form acrylamide, the differences between the varieties were not large enough to obtain a sufficiently low level of this harmful substance when choosing another variety. Of course, study that do not give breakthrough results, but in case it describes completely new raw materials, new methods, etc. In this case, the research itself is not innovative, and combined with the lack of favorable results. However, this is not the only drawback of this manuscript.
Reading the following fragments of the manuscript, it is hard to resist the impression that the authors have studied the bibliography rather cursily and approach their experiences partly uncritically and partly with unreasonable enthusiasm.
And so as below:
Figure 1: The title is not very precise, it says nothing about what the filling is about. Regardless of the degree of filling, the rye was roasted for 13 minutes, the grain temperature was not measured, and it is known that the larger the filling, the easier the grain heats and roasts faster from the moment the process becomes exothermic, so the differences in this case are simply roasting intensity differences (the mass and the color of beans roasted in different conditions differ, so how to compare the samples). If the authors reported what the grain color was after roasting in each of the studied cases of filling the roaster and the temperature of the grains, not of the roasting air, then the situation would be clearer. It is not written which variety was roasted in the experiment regarding the level of filling of the roaster.
Figure 4: Here again a similar remark, the authors compared roasted rye at different temperatures, but at the same time of roasting. Therefore, at 120 C degrees they obtained a much lower content of acrylamide, the grain was probably much brighter, so comparing it with grain roasted at other temperatures and in the equal time is unreasonable. In addition, presenting results in such a way that the continuity of temperature rise is not maintained on the axis deviates from generally accepted recommendations.
Adv Exp Med Biol. 2005;561:317-28. doi: 10.1007/0-387-24980-X_24.
J Agric Food Chem. 2006 Oct 4;54(20):7847-55. doi: 10.1021/jf0611264.
Figure 5a: The authors describe as a new achievement the decreasing acrylamide content with prolonged roasting, while at such a high temperature and in such a long time it is proven that the acrylamide formed is further decomposing and volatilizing during roasting and a certain maximum concentration is observed during roasting in quite drastic conditions.
Figure 5c: The result of acrylamide content in roasted rye depending on the pH of the grain appears to be unjustified in collision with literature data, e.g., which says that low pH reduces the reaction rate of acrylamide formation.
Figure 7: Gradual darkening of the grains with increasing roasting temperature is a well-known fact. The authors did not capture in their results more detailed, but also known facts, which explain the observed color stabilization when roasting at a temperature between 170 and 190 C degrees. Well, these results relate to roasting, in which as a result of pyrolysis a significant amount of water was emitted at the end of roasting, which limited the roasting progress in this temperature range.
Bibliography:
Some publication titles are not written in accordance with the journal's requirements, unnecessarily every word with capital letter.
Citations did not include publications:
Talanta. 2011 Oct 30;86:23-34. doi: 10.1016/j.talanta.2011.08.066. Epub 2011 Sep 19.
Author Response
The manuscript describes the usefulness of roasted rye as a substitute for roasted coffee. Such an idea resulted from the market demand for coffee-like products due to the needs of consumers who do not want to consume caffeine. Currently, grains of other cereals are used for this purpose, e.g. of barley or wheat, roasted chicory is also used. Currently used roasted substitutes have a high content of acrylamide, whose presence in food should be limited. Hence the search for other substitutes.
The study uses rye that has undergone various modifications before or during roasting. None of the modifications allowed to achieve the level of acrylamide lower than recommended in the EU. Even if we notice that the authors chose for the study a variety especially predisposed to form acrylamide, the differences between the varieties were not large enough to obtain a sufficiently low level of this harmful substance when choosing another variety. Of course, study that do not give breakthrough results, but in case it describes completely new raw materials, new methods, etc. In this case, the research itself is not innovative, and combined with the lack of favorable results. However, this is not the only drawback of this manuscript.
Reading the following fragments of the manuscript, it is hard to resist the impression that the authors have studied the bibliography rather cursily and approach their experiences partly uncritically and partly with unreasonable enthusiasm.
We thank the reviewer for critically reading our manuscript. Indeed, there was room for improvement and clarification, accordingly, we tried to address the raised points properly.
- Figure 1: The title is not very precise, it says nothing about what the filling is about. Regardless of the degree of filling, the rye was roasted for 13 minutes, the grain temperature was not measured, and it is known that the larger the filling, the easier the grain heats and roasts faster from the moment the process becomes exothermic, so the differences in this case are simply roasting intensity differences (the mass and the color of beans roasted in different conditions differ, so how to compare the samples). If the authors reported what the grain color was after roasting in each of the studied cases of filling the roaster and the temperature of the grains, not of the roasting air, then the situation would be clearer. It is not written which variety was roasted in the experiment regarding the level of filling of the roaster.
We adapted the title as suggested and added information regarding rye variety and grain colour.
- Figure 4: Here again a similar remark, the authors compared roasted rye at different temperatures, but at the same time of roasting. Therefore, at 120 C degrees they obtained a much lower content of acrylamide, the grain was probably much brighter, so comparing it with grain roasted at other temperatures and in the equal time is unreasonable. In addition, presenting results in such a way that the continuity of temperature rise is not maintained on the axis deviates from generally accepted recommendations.
The reviewers assumption regarding grain brightness is correct. Therefore, we evaluated brightness (Figure 7) and discussed consumer acceptance in the discussion section. Brightness values of <65 are typically required which cannot be met by samples roasted at 120 °C. Figure 4 was adapted according to reviewers recommendations.
- Figure 5a: The authors describe as a new achievement the decreasing acrylamide content with prolonged roasting, while at such a high temperature and in such a long time it is proven that the acrylamide formed is further decomposing and volatilizing during roasting and a certain maximum concentration is observed during roasting in quite drastic conditions.
We changed the respective paragraph in both, results and discussion sections. It was not our intention to present the decreased acrylamide content as a new achievement, but rather perform it as a control experiment.
- Figure 5c: The result of acrylamide content in roasted rye depending on the pH of the grain appears to be unjustified in collision with literature data, e.g., which says that low pH reduces the reaction rate of acrylamide formation.
The reviewer’s opinion that lower pH of grains should further reduce the acrylamide content makes sense. However, we did not detect this effect in our study and addressed this point in the discussion section. We believe that acetic acid, which was used for acidification, was lost during the roasting process.
- Figure 7: Gradual darkening of the grains with increasing roasting temperature is a well-known fact. The authors did not capture in their results more detailed, but also known facts, which explain the observed color stabilization when roasting at a temperature between 170 and 190 C degrees. Well, these results relate to roasting, in which as a result of pyrolysis a significant amount of water was emitted at the end of roasting, which limited the roasting progress in this temperature range.
We thank the reviewer for this very valuable hint including a possible explanation. We rechecked our data. Stabilization of sample brightness in the mentioned temperature range is in fact observable. We added a short paragraph in the discussion section.
- Bibliography:
Some publication titles are not written in accordance with the journal's requirements, unnecessarily every word with capital letter.
The literature included in our bibliography was added with their original title using EndNote. To our knowledge there are no specific requirements regarding capital letters in the title.
- Citations did not include publications:
Talanta. 2011 Oct 30;86:23-34. doi: 10.1016/j.talanta.2011.08.066 . Epub 2011 Sep 19.
Paper was added as requested in the introduction section.
Reviewer 2 Report
Dear authors,
the manuscript describes the reduction of acrylamide for roasted rye, a possible substitute of coffee. Several patents and previous work are found in the current theme, nevertheless, I found the work interesting and very well written.
Therefore, please revise carefully all the manuscript for some spelling English errors that you might find. Also, I propose to transform some methodology (2.2) in a flowchart of steps (coupled to a smaller description), or separate by section each extraction type. Please, turn that section more appealing.
The results and discussion are appropriate, the bibliography is also appropriate.
I advise making the proposed changes to improve the overall quality of the manuscript.
Author Response
The manuscript describes the reduction of acrylamide for roasted rye, a possible substitute of coffee. Several patents and previous work are found in the current theme, nevertheless, I found the work interesting and very well written.
Therefore, please revise carefully all the manuscript for some spelling English errors that you might find. Also, I propose to transform some methodology (2.2) in a flowchart of steps (coupled to a smaller description), or separate by section each extraction type. Please, turn that section more appealing. The results and discussion are appropriate, the bibliography is also appropriate. I advise making the proposed changes to improve the overall quality of the manuscript.
We thank the reviewer for critically reading the manuscript and for the positive feedback. The manuscript was carefully revised for spelling errors. As suggested, the methodology section (2.2.) was separated into 8 subchapters for better readability.
Reviewer 3 Report
The manuscript by Pitsch et al. reports the applicability of roasted rye as a coffee substitute. Cofee substitutes are non-coffee products, such as roasted barley, malted barley, chicory and rye.
It can hardly be accepted that roasted rye can be applicable as a coffee substitute, as it has not caffeine on its composition. This is a non-coffee product.
"To investigate the applicability of roasted rye a as a coffee substitution product, rye grains of a single type were roasted in a drum roaster." The experimental design does not make it possible to conclude that the roasted rye can be applied as a coffee substitution.
The manuscript is well written, and the overall experimental approach is appropriate.
My only concerns goes to the title, and the inclusion of the term "cofee substitute". In addition, no reference is made to the coffee properties in the introductory section of this manuscript.
Author Response
The manuscript by Pitsch et al. reports the applicability of roasted rye as a coffee substitute. Cofee substitutes are non-coffee products, such as roasted barley, malted barley, chicory and rye. It can hardly be accepted that roasted rye can be applicable as a coffee substitute, as it has not caffeine on its composition. This is a non-coffee product.
"To investigate the applicability of roasted rye a as a coffee substitution product, rye grains of a single type were roasted in a drum roaster." The experimental design does not make it possible to conclude that the roasted rye can be applied as a coffee substitution.
The manuscript is well written, and the overall experimental approach is appropriate.
- My only concerns goes to the title, and the inclusion of the term "cofee substitute". In addition, no reference is made to the coffee properties in the introductory section of this manuscript.
We thank the referee for reviewing our paper and the positive feedback. Regarding the title, we disagree that roasted rye cannot be applicable as a coffee substitute. As mentioned by the reviewer “Cofee substitutes are non-coffee products, such as roasted barley, malted barley, chicory and rye“. Furthermore, caffeine-free products defined as “coffee substitutes” are common in science (https://doi.org/10.3390/molecules24183377) and popular science (https://en.wikipedia.org/wiki/Coffee_substitute). Therefore, we decided to keep the title as used in previous submission.
As requested, we added a reference regarding coffee properties of roasted rye.
Round 2
Reviewer 1 Report
The manuscript can be published in the current version.
Reviewer 3 Report
The manuscript has been substantially improved and can be accepted for publication in its current form.